# Overexpression of Hepatocyte Chemerin-156 Lowers Tumor Burden in a Murine Model of Diethylnitrosamine-Induced Hepatocellular Carcinoma

**DOI:** 10.3390/ijms21010252

**Published:** 2019-12-30

**Authors:** Elisabeth M. Haberl, Rebekka Pohl, Lisa Rein-Fischboeck, Susanne Feder, Christopher J. Sinal, Astrid Bruckmann, Marcus Hoering, Sabrina Krautbauer, Gerhard Liebisch, Christa Buechler

**Affiliations:** 1Department of Internal Medicine I, Regensburg University Hospital, 93053 Regensburg, Germany; Haberl.elisabeth@gmx.de (E.M.H.); becky-pohl@web.de (R.P.); lisa.rein-fischboeck@gmx.de (L.R.-F.); susanne.feder@klinik.uni-regensburg.de (S.F.); 2Department of Pharmacology, Dalhousie University, Halifax, NS B3H 4R2, Canada; Christopher.Sinal@Dal.Ca; 3Biochemistry Center Regensburg (BZR), Laboratory for RNA Biology, University of Regensburg, 93042 Regensburg, Germany; astrid.bruckmann@vkl.uni-regensburg.de; 4Institute of Clinical Chemistry and Laboratory Medicine, Regensburg University Hospital, 93053 Regensburg, Germany; marcus.hoering@klinik.uni-regensburg.de (M.H.); sabrina.krautbauer@klinik.uni-regensburg.de (S.K.); gerhard.liebisch@klinik.uni-regensburg.de (G.L.)

**Keywords:** Triglycerides, chemokine-like receptor 1, chemerin activity, liver, adenoassociated virus

## Abstract

The tumor inhibitory potential of the highly active chemerin-156 isoform was described in orthotopic models of hepatocellular carcinoma (HCC). The majority of HCC arises in the fibrotic liver, which was not reproduced in these studies. Here, a potential therapeutic activity of chemerin-156 was evaluated in diethylnitrosamine (DEN)-induced liver cancer, which mimics fibrosis-associated HCC. Mice were infected with adeno-associated virus (AAV) six months after DEN injection to overexpress chemerin-156 in the liver, and animals injected with non-recombinant-AAV served as controls. Three months later, the animals were killed. Both groups were comparable with regard to liver steatosis and fibrosis. Of note, the number of very small tumors was reduced by chemerin-156. Anyhow, the expression of inflammatory and profibrotic genes was similar in larger tumors of control and chemerin-156-AAV-infected animals. Although genes with a role in lipid metabolism, like *3-hydroxy-3-methylglutaryl-coenzym-A*-*-reductase*, were overexpressed in tumors of animals with high chemerin-156, total hepatic cholesterol, diacylglycerol and triglyceride levels, and distribution of individual lipid species were normal. Chemerin-156-AAV-infected mice had elevated hepatic and systemic chemerin. Ex vivo activation of the chemerin receptor chemokine-like receptor 1 increased in parallel with serum chemerin, illustrating the biological activity of the recombinant protein. In the tumors, chemerin-155 was the most abundant variant. Chemerin-156 was not detected in tumors of the controls and was hardly found in chemerin-156-AAV infected animals. In conclusion, the present study showed that chemerin-156 overexpression caused a decline in the number of small lesions but did not prevent the growth of pre-existing neoplasms.

## 1. Introduction

Hepatocellular carcinoma (HCC) is among the deadliest solid cancers, with the main etiologies being viral infections and non-alcoholic steatohepatitis (NASH) [1]. Chronic liver injury and HCC progression are characterized by inflammation, regenerative processes, and liver fibrosis [2]. Based on experimental evidence indicating a role of myeloid cells in supporting tumor angiogenesis, metastasis, and progression, the dysregulated response of immune cells is believed to contribute to tumor growth in HCC [2,3]. Thus, strategies to antagonize the tumor-promoting activities of myeloid cells may reduce tumor burden in HCC [3].

The chemoattractant protein chemerin is involved in inflammation, and regulates the recruitment and function of innate and adaptive immune cells [4]. Chemerin is produced primarily by adipocytes and hepatocytes, and is secreted in a pro-form that is subsequently activated by C-terminal proteolysis [4,5]. Several chemerin isoforms are generated by this processing, with murine chemerin-156 and human chemerin-157 having the greatest chemoattractant activity for macrophages expressing the chemerin receptor chemokine-like receptor 1 (CMKLR1) [6].

Reduced chemerin expression and an anti-tumor effect for chemerin have been reported for several forms of cancer [7]. For example, chemerin expression is low in adrenocortical carcinoma and chemerin overexpression in immune-deficient mice reduced tumor growth. This was in line with demonstrated in vitro inhibitory effects on cell proliferation, invasion, and tumorigenicity [8]. Mechanistically, this was attributed to a direct chemerin-dependent increase in the degradation of β-catenin and an impaired phosphorylation of p38 mitogen-activated protein kinase in tumor cells [8]. Other anti-tumor effects of chemerin have been attributed to alterations in immune function. For example, the growth inhibitory activity of chemerin in a murine melanoma model is associated with an increased number of natural killer cells and the depletion of myeloid-derived suppressor cells and plasmacytoid dendritic cells [9]. In contrast to these anti-cancer effects, neuroblastoma tumor growth is reportedly reduced when chemerin/CMKLR1 signaling is blocked [10]. Moreover, in squamous cell carcinoma of the oral tongue, high chemerin expression is correlated with a poorer patient outcome [11] and further tumor-promoting effects of chemerin were identified in gastric cancer and squamous esophageal cancer cells [12,13].

In human HCC tissues, chemerin protein expression is low in comparison to non-tumorous liver tissues. Tumor chemerin protein levels are an independent prognostic factor and are inversely associated with tumor grade and size. Positive correlations with the number of dendritic and natural killer cells have indicated an immune-regulatory role of chemerin in HCC [14]. Accordingly, a protective function of chemerin was proposed in an orthotopic murine HCC model. Consistent with this, chemerin overexpression blocked aggressive tumor growth and metastasis in chemerin knock-out mice. This was attributed to reduced activation of nuclear factor-κB, as well as the expression of granulocyte-macrophage colony-stimulating factor and IL-6. This was accompanied by a decline of myeloid-derived suppressive cells and a concomitant increase of interferon-γ^+^ T cells [15]. A separate study showed that chemerin inhibited migration, invasion, and metastasis of HCC cells via disruption of the CMKLR1/phosphatase and tensin homolog (PTEN) complex, allowing PTEN to exert its tumor suppressor activities [16].

One disadvantage of xenograft models is the considerable differences between cell lines, and the use of multiple cell lines is recommended [17]. Moreover, most primary liver tumors arise in the cirrhotic liver and the therapeutic effect of chemerin during fibrosis-associated carcinogenesis cannot be tested by the use of xenograft models [1]. For this purpose, the diethylnitrosamine (DEN)-induced HCC model is suited. DEN injection causes DNA damage, and later on, oxidative stress, steatosis, and fibrosis develop in the liver [17,18,19,20]. This model is supposed to reproduce human HCC with poor prognosis [18]. Different studies analyzed hepatocarcinogenesis in the DEN model. Premalignant lesions were induced 24 weeks after DEN injection and tumors were easily detected three months later [21,22,23,24]. Therefore, chemerin was overexpressed in the liver of mice 24 weeks after DEN application. It is important to note that disease progression from 24 to 40 weeks was mostly because of the growth of preexisting lesions, whereas tumor number, at most, doubled [23,24,25,26]. Chemerin-156 is a highly active murine isoform and was analyzed in previous studies illustrating anti-cancer effects in HCC [15,16,27]. The activity of this isoform has not been studied in fibrosis-associated HCC until now. Moreover, chemerin-156 abundance in the liver is still unknown. Here, we investigate the effect of chemerin-156 in the DEN model. Active chemerin is overexpressed at an early stage of the disease until the end of the experiment, where tumors are detected in the liver. Chemerin-156 reduces the number of small tumors but cannot prevent the progression of pre-existing lesions to HCC.

## 2. Results

### 2.1. Body Composition of Chemerin-156-Expressing Mice

Previously, chemerin-156 overexpression was shown to inhibit liver tumor growth in an orthotopic mouse model [15]. In the present study, a potential therapeutic activity of this chemokine was evaluated in the DEN-induced HCC model. As shown in Figure 1a, very young C3H/HeNRj mice were injected with DEN and 24 weeks later, control or chemerin-156-producing AAV particles (1 × 10^12^) were administered into the peritoneal cavity. Thirteen weeks later, the mice were killed. Of note, the body weight of both groups was similar during the experiment (Figure 1b). Chemerin-156 overexpression had no effect on total body weight and did not impact on subcutaneous, epididymal, and perirenal fat pad weights (Figure 1b–d, median perirenal fat pad weight of controls was 7.6 mg/g and 6.8 mg/g for chemerin-156 expressing mice). Similarly, the spleen weight was comparable for both groups (median spleen weight of controls was 3.6 mg/g and 3.3 mg/g for chemerin-156 expressing mice, as was the liver to body weight ratio (Figure 1e). Comparable body and organ weights indicate minor, if any, differences in liver injury, tumor number, and size. Of note, the liver to body weight ratio was negatively correlated with epididymal and perirenal fat to body weight ratios (Figure 1f,g). This was also observed when both groups were analyzed separately. There was also a modest negative correlation with subcutaneous fat in the whole group (*r* = −0.480, *p* = 0.018).

### 2.2. Serum and Hepatic Chemerin Protein and Activity of Serum Chemerin

Serum chemerin was measured immediately before and 1, 4, 8, 12, and 13 weeks after AAV injection. Total chemerin protein was higher at all the time points for chemerin-156-AAV-infected mice (Figure 2a). Chemerin activity in serum was measured at the end of the study. The ex vivo activation of CMKLR1 was higher in chemerin-156-infected mice, whereas the activation of G protein-coupled receptor 1 (GPR1) by serum chemerin was not significantly induced (Figure 2b,c). Hepatic chemerin protein was about two-fold increased in chemerin-156-AAV-infected mice (Figure 2d). Overall, these data confirm raised hepatic production and release of chemerin into the circulation.

### 2.3. Tumor Number and Size

Macroscopic examination of the livers of control-AAV and chemerin-156-AAV infected mice did not reveal gross differences (Figure 2e). However, mice with chemerin-156 overexpression had about 30% fewer tumors in the liver (Figure 2f). When stratified for size (<1, 1–2, 2–5, 5–10, >10 mm), tumor number was lower in each category. Tumors with a diameter smaller than 1 mm were significantly reduced (Figure 2g). This illustrates that chemerin-156 overexpression reduced tumor burden. With regard to previous studies, which showed that disease progression was mostly due to the growth of preexisting lesions [23,24,25,26], it is likely that chemerin-156 hindered the formation of neoplasm. The number of larger tumors did not significantly differ between the two groups. These tumors did most likely arise from neoplastic lesions already present at the time of AAV application. Thus, chemerin-156 did not prevent tumor growth of existing lesions. Accordingly, the ratio of tumors with a diameter <2 mm to tumors with a diameter ≥ 2 mm was identical in both groups (Figure 2h).

Alpha-fetoprotein (AFP) is an accepted serum biomarker for HCC [28]. When measured at the time of AAV injection and 1, 4, 8, 12, and 13 weeks later, AFP increased during disease progression but was comparable in chemerin-156 and control-AAV injected mice at all time points (Figure 2i). AFP is insensitive for the detection of small liver tumors [28], and thus, a reduced number of small tumors in chemerin-156-expressing mice did not translate to lower AFP.

### 2.4. Markers of Liver Injury

Liver fibrosis is a risk factor for HCC and the application of DEN induces liver steatosis and fibrosis [19,20]. Liver histology by hematoxylin and eosin staining revealed no differences between the two groups (Figure 3a). The extent of liver steatosis appeared comparable, and accordingly, levels of hepatic triglycerides and cholesterol were similar (Figure 3a–c). Ceramides contribute to liver steatosis and fibrosis and the biosynthesis of these lipids was enhanced by DEN [20,29]. Ceramide concentrations did not differ between the animal groups (Figure 3d). The normal range of the hepatic phosphatidylcholine (PC)/phosphatidylethanolamine (PE) ratio is between 1.5 and 2.0, and higher as well as lower ratios were linked to liver disease [30]. The PC/PE ratio was similar in both groups, indicating that chemerin-156 overexpression did not modulate liver injury induced by DEN (Figure 3e). Sirius red staining showed a comparable degree of liver fibrosis in mice with chemerin-156 overexpression and the respective control animals (Figure 3f). Likewise, *α-smooth muscle actin* (*α-SMA*) and *collagen* (*Col)4a3* mRNA were similarly expressed in the non-tumorous liver of both groups (Figure 4a,b). These findings clearly show that the reduced tumor burden of mice with chemerin-156 overexpression was not related to improved liver function.

### 2.5. Genes and Proteins Already Described to Be Differentially Expressed in Cancer

As remodeling of the extracellular matrix is required for tumor progression [31], the expression of several genes involved in this process was measured. The expression of *α-SMA* and *Col4a3* mRNA was higher in the tumorous than non-tumorous tissues of all mice, regardless of chemerin-156 overexpression (Figure 4a,b). Consistent with previous reports [32,33,34,35], *early growth response gene-1* (*Egr-1*), *solute carrier family 12 member 1* (*Slc12a1*), and *serine peptidase inhibitor, Kazal type 1* (*Spink*1) mRNA levels were higher in tumorous than non-tumorous tissues, whereas *glucose-6-phosphatase* (*G6PC)* was reduced (Figure 4c–f). However, this effect was similar regardless of chemerin-156 overexpression.

The activation of β-catenin was commonly described in HCC [36]. Indeed, mRNA expression of this gene was non-significantly induced in HCC tissues of both mice groups (Figure 4g). Protein levels of β-catenin were not higher in the tumors and did not differ between the groups (Figure 4h,i). Phosphorylation of β-catenin at S552 by Akt induces nuclear translocation of β-catenin [37], whereas phosphorylation of β-catenin at T41, S37, and S33 initiates its degradation [36]. Analysis of these phosphorylated β-catenins showed no difference between the mice with hepatic expression of chemerin-156 and controls (Figure 4h,j,k). Moreover, the abundance of these isoforms was not changed in the cancer tissues (Figure 4h,j,k). Mitogen-activated protein kinases like p38 contribute to HCC development [38], but *p38* mRNA levels were changed neither in the tumors nor by chemerin-156 overexpression (Figure 4l). Accordingly, it was shown by others that p38 protein and its phosphorylated form were not altered in tumors of DEN-injected mice [39].

### 2.6. Analysis of Genes Highly Expressed by Macrophages and Natural Killer Cells

Chemerin is an established chemoattractant for immune cells. Thus, the expression of several pro-inflammatory genes (*F4/80*, *CD38*, *IL-6*) and genes characteristic for natural killer cells (*NCR1*, *Ly49c*) was also analyzed. The mRNA level of these genes was comparable in tumorous and non-tumorous tissues for both groups (Table 1).

Thus, while the differential expression of several genes in tumorous tissue reported in other HCC studies was reproduced herein, there were no apparent differences between control-AAV- and chemerin-156-AAV-infected mice with respect to these endpoints (Figure 4 and Table 1). In order to broaden the scope of this investigation, global gene expression analysis was performed. There were 224 genes, which were upregulated more than two-fold in tumorous compared to non-tumorous tissue of the control animals (Appendix A). Of these genes, 223 were similarly induced in tumorous tissues of control-AAV- and chemerin-156-AAV-infected mice (Appendix A). The sole exception was *Gm25482*, a predicted gene with unknown function, which was less abundant in the tumors of chemerin-156-overexpressing mice (Appendix A). Moreover, the established HCC markers [40], *AFP* (11.4- and 12.3-fold higher in the tumors of control and chemerin-156 infected mice, respectively) and *glypican-3* (17.3- and 21.4-fold higher in the tumors of control and chemerin-156 infected mice, respectively) were markedly increased in tumorous versus non-tumorous tissues for both groups (Appendix A).

Microarray analysis further identified 105 genes to be more than two-fold reduced in tumorous versus non-tumorous tissues of control-AVV-infected animals (Appendix A). However, none of these genes differed in the tumors of control-AVV- and chemerin-156-AAV-infected mice. Taken together, the overall pattern of differential gene expression was in agreement with similar tumor malignancies in both groups. The cluster dendrogram accordingly showed common gene expression patterns in tumors and paratumorous tissues of both mice groups (Appendix A). Principal component analysis separated HCCs and non-tumorous tissues but not control- and chemerin-156-infected mice (Figure 4m).

Intriguingly, the *dedicators of cytokinesis 8* (*DOCK8*; fold change: 1.19), *phospholysine phosphohistidine inorganic pyrophosphate phosphatase* (*LHPP*; fold change: 1.63), and *cytochrome P450, family 1, subfamily a, polypeptide 2* (*CYP1A2*; fold change: 1.19) were high in the non-tumorous liver tissues of chemerin-156 compared to control-AVV-infected mice (Appendix A). The mRNA levels of these genes were all reduced in tumor tissues, a finding consistent with human HCC [41,42,43]. Thus, higher expression of these genes in the normal liver of mice with chemerin-156 overexpression may contribute to a delay in tumorigenesis or may simply indicate a lower number of precancerous lesions.

### 2.7. Lipid Metabolism in Tumorous and Non-Tumorous Liver Tissue

Reprogramming of lipid metabolism is fundamental for rapidly proliferating tumor cells [44]. This led us to analyze the expression of genes and proteins with a function in lipid metabolism. *3-hydroxy-3-methylglutaryl-coenzym-A -reductase* (*HMG-CoA-R*) mRNA was significantly higher in tumorous versus non-tumorous tissues for both groups and was most highly expressed in tumor tissues from chemerin-156-overexpressing mice (Figure 5a, Appendix A). Apolipoprotein A1 (ApoA1) is the main apolipoprotein of high-density lipoprotein. Both *ApoA1* mRNA and protein levels were similarly reduced in the tumors of both groups (Figure 5b,c and Appendix A). Fatty acid binding protein 5 (*Fabp5*) mRNA and protein levels were increased in the tumorous versus non-tumorous tissues of the chemerin-156-overexpressing mice, but not the control group. However, while tumor *Fabp5* mRNA levels were significantly greater for chemerin-156-overexpressing mice, tumor Fabp5 protein levels were similar for both groups (Figure 5d–f and Appendix A). *Arachidonate 5-lipoxygenase* (*Alox5*) mRNA was significantly higher in tumor tissue and did not differ between treatment groups (Figure 5g and Appendix A). *Patatin-like phospholipase domain containing 5* (*Pnpla5*) mRNA levels were markedly higher in the tumors of chemerin-156-, but not control-AVV-infected mice (Figure 5h and Appendix A). Protein levels of full-length and proteolytic activated sterol regulatory element binding protein (SREBP) 1c and SREBP2, of stearoyl-CoA-reductase 1 (SCD1), of fatty acid synthase (FAS), and Staphylococcal nuclease domain-containing protein 1 (SND1) were not different between tumorous and non-tumorous tissue and were not affected by chemerin-156 overexpression (Appendix A and Figure 5i).

*HMG-CoA-R* is a central enzyme in cholesterol synthesis, whereas *Pnpla5* has neutral lipid triacylglycerol lipase and acylglycerol transacylase activity [45,46]. Higher expression of these genes in tumors of chemerin-156-expressing mice led us to perform lipidomic analysis of liver tumors and non-tumorous tissue. Levels of total cholesterol, triglycerides, and diacylglycerols, as well as triglyceride to diacylglycerol ratios were higher in the tumorous versus non-tumorous tissue of all mice, but did not differ between control-AVV and chemerin-156-AAV groups (Figure 6a–d). Analysis of 52 individual triglyceride species showed increased levels for all in the tumors of both groups (Appendix A). Of the 18 analyzed diacylglycerol species, 15 were also higher in tumors (Appendix A). However, the levels of these lipids in tumor and non-tumorous tissue were not changed by chemerin overexpression (Appendix A). Lipid analysis thus excludes an effect of chemerin-156 in the progression of precursor nodules or cancer malignancy.

### 2.8. Chemerin and CMKLR1 in Murine HCC

Previous studies described low chemerin in human HCC [7]. In murine NASH livers, chemerin protein was highly expressed and was not changed in the tumors [47]. Chemerin protein was also determined in tumorous and non-tumorous liver tissues of the DEN mouse model. Overall, chemerin was highest in tissues from chemerin-156 infected mice (Figure 7a–c). Chemerin protein tended to be higher in the HCC tissues of control-AAV infected mice and was not changed in tumors of chemerin-156-AAV injected animals (Figure 7a,b).

Chemerin mRNA was highest in the normal liver tissue of mice overexpressing chemerin-156. A decline in chemerin mRNA was observed in tumor tissues of these mice, but not in the control-AAV-injected controls (Figure 7c). Protein levels of CMKLR1 were comparably reduced in the tumor tissues of both groups (Figure 7d–e). CMKLR1 mRNA was not changed by either chemerin-156 overexpression or in the tumors (Figure 7f).

Hepa1-6 cells express low levels of chemerin and were used to overexpress different chemerin isoforms. The cells transfected with the plasmids to express the isoforms chemerin-162, 156, 155, and 154 had similar chemerin protein in the respective lysates (Figure 7g). This shows that the chemerin antibody used herein binds to all of these isoforms and excludes a marked preference for a specific chemerin variant. Thus, in contrast to human findings, hepatic chemerin protein was not changed in murine HCC.

### 2.9. Chemerin Isoforms in Murine Liver Tumors

Pro-chemerin undergoes C-terminal proteolytic processing to produce several isoforms that vary in biological activity. Chemerin isoforms have been analyzed in serum and adipose tissue but not in the liver, so far [27]. Therefore, we decided to identify the chemerin isoforms in the hepatic tumor tissues. Chemerin-155 was detected in all of the tumors analyzed. Chemerin-154 and chemerin-153 were present in about 50% of the liver tumors, and the latter variant was found more often in mice with chemerin-156 overexpression. Of note, chemerin-156 was not found in tumors of control-AAV-injected animals. This isoform was detected in 25% of the liver tumors of animals overexpressing chemerin-156 (Figure 7h). This demonstrates that chemerin-156 is not an endogenous isoform of murine hepatic tumors.

## 3. Discussion

The present study suggested that hepatic overexpression of chemerin-156 in early hepatocarcinogenesis hindered the formation of neoplasms but not the growth of preexisting lesions. Moreover, chemerin-156 was not detectable in the liver tumors of control-AVV-infected mice. These data challenge the current vision that the apparent beneficial relationship of total hepatic chemerin protein with HCC prognosis is related to the chemerin-156 isoform. In contrast, our data indicate that chemerin-155 was highly abundant in the murine liver tumors, warranting future studies to evaluate the role of this isoform in liver tumorigenesis.

Recent studies described protective effects of chemerin-156 in HCC models. Implantation of subcutaneous grown tumor tissues derived from mouse HCC cell lines into the liver was one of the models studied [15]. A separate analysis injected HCC cells mixed with matrigel into the liver of nude mice [16]. Both studies described markedly reduced tumor burden upon chemerin-156 overexpression or injection [15,16].

In the DEN model analyzed herein, fewer tumors were present when chemerin-156 was overexpressed. This is in general accordance with the protective role of chemerin-156 described in previous studies [15,16]. However, in the present model, the primary effect was to reduce the number of very small tumors. Analysis of gene and protein expression and measurement of various lipid species in the larger tumors did not identify any gross differences between control-AVV- and chemerin-156-AAV-injected mice. HCC progresses from hyperplastic nodules to adenomas to carcinomas [22,23]. At the time of chemerin-156 overexpression, mice had already developed preneoplastic lesions [22,23]. Larger tumors originate from these hyperplastic nodules. Chemerin-156 had no effect on tumor progression. The number of large tumors and the degree of tumor malignancy did not differ between the two groups of animals. Cancer-associated fibroblasts within the HCC environment contribute to disease progression. These cells express α-SMA, which is associated with poor survival of patients with HCC [48]. In the tumors, *α-SMA* was comparably induced in both groups of mice in accordance with similar malignancy of liver tumors. The HCC biomarker AFP similarly increased during disease progression in all of the mice, further illustrating comparable tumor growth. Small tumors and neoplastic lesions usually do not secrete AFP and do not affect its serum level [28].

The mechanisms by which chemerin may prevent formation of liver lesions remains unknown. Liver fibrosis and bioactive lipids like ceramides contribute to the pathogenesis of liver tumors [1,44]. Based on histological, gene expression, and lipidomic data, chemerin-156 did not improve liver function. Cancer is associated with adipose tissue loss, but fat pad weights were not changed by chemerin overexpression. Of note, there was a negative correlation of liver to body weight ratio and intraabdominal fat pad weights. Fat atrophy appears to be triggered by the tumor and may supply cancers with fatty acids to generate ATP [49]. Overexpression of chemerin in the liver did not alter adipose tissue mass and appears not to interfere with energy supply.

Chemically-induced liver tumorigenesis is a stepwise process with distinct stages of initiation, promotion, and progression [50]. The current model indicates that chemerin-156 retards initiation and/or early tumor growth. The expression of three genes, *DOCK8*, *LHPP*, and *CYP1A2*, was induced in the normal liver tissues of chemerin-156-AVV compared to control-AAV infected mice. The suppression of these genes in the murine tumors and human HCC [41,42,43] suggests a role in hepatocarcinogenesis. Future experiments have to evaluate a potential anti-tumor capacity of these proteins. Again, equal expression of these genes in the HCC of both animal groups showed that chemerin-156 had little or no protective role in the progression of HCC. Principally, this seems to contradict recent studies demonstrating the anti-cancer effects of this chemerin isoform in xenograft models [15,16].

Compared to DEN-induced HCC, xenograft models do not develop liver fibrosis [51]. Proteinases like chymase were induced in the fibrotic liver and may contribute to chemerin inactivation [52,53]. Chromosomal aberrations, inter- and intra-tumor heterogeneity are prevalent in carcinogen-induced hepatocarcinogenesis [51] and affect cell function. The heterogeneous sensitivity of HCC cells to drugs like sorafenib has been documented [54] and the responsiveness of different HCCs to chemerin-156 may vary. Furthermore, CMKLR1 was strongly downregulated in the HCC tissues of the mice studied herein, and this may further abolish the protective effects of chemerin-156. However, CMKLR1 levels were maintained in the tumors of mice 21 weeks after DEN injection [47], consistent with the ability of chemerin-156 to exert beneficial activities in early carcinogenesis.

Chemerin expression was analyzed in different human cancers, whereas levels of CMKLR1 were hardly investigated. The human protein atlas provides information about CMKLR1 in colorectal, breast, prostate, lung, and thyroid cancer [55]. CMKLR1 protein was not prognostic in these different tumor diseases [55]. In the liver, CMKLR1 is expressed by hepatocytes, Kupffer cells, bile-duct cells, and hepatic stellate cells [56]. Preliminary data from our group suggest that CMKLR1 protein was reduced in non-viral human HCC [57]. Until now, the factors regulating CMKLR1 in liver cells have hardly been described. Adiponectin and IL-6 seem to induce CMKLR1 mRNA in hepatocytes. In hepatitis C infection, hepatic CMKLR1 was normal, levels were reduced or induced in NASH [5]. Further research has to analyze CMKLR1 expression in HCC as well as chronic liver diseases.

The contribution of serum chemerin to HCC is a further unresolved question. Patients with liver cirrhosis are at risk of developing HCC [58]. Low systemic chemerin was associated with mortality in patients with decompensated disease [59]. On the other hand, serum chemerin was comparable in non-cancer patients, patients with colorectal liver metastases, and HCC [60]. Most chemerin in serum is inactive [61], and detailed analysis of chemerin isoforms in patients with cirrhosis and HCC may help to resolve this issue.

Full-length chemerin overexpression in adrenocortical carcinoma cells promoted β-catenin degradation, which was initiated by phosphorylation of the protein at S33, S37, or T41. Moreover, it inhibited the phosphorylation of p38 kinase [8]. Subsequently, cell proliferation, invasion, and tumorigenesis were blocked. Interestingly, exogenously-provided active chemerin-157 had no effect on these features [8]. Aberrant activation of the Wnt/β-catenin pathway was reported in liver cancer, but nuclear localization of β-catenin was not detected in the DEN model [60]. Mice with hepatocytes unable to secrete Wnts had comparable tumor burden and HCC histology as the respective controls [62]. These studies showed that β-catenin was dispensable during DEN-induced HCC [62,63]. Indeed, protein levels of β-catenin were essentially identical in tumorous and non-tumorous tissues, and did not vary between the groups. Moreover, p38 protein and its phosphorylated form were not changed in tumors of DEN-injected mice [39]. Accordingly, *p38* mRNA was not upregulated in the tumors nor by chemerin-156 overexpression. Akt was activated by chemerin in short-term exposure. Treatment with chemerin for a longer time caused a decline in the levels of phosphorylated Akt. Of note, after 4 h incubation, Akt activation reverted to the resting levels [16]. There was no evidence to suggest that Akt was activated by chemerin-156 in the cancer tissues analyzed. This kinase affects various signaling pathways and nearly identical gene expression in tumor and paratumor tissues of chemerin-156-AAV- and control-AAV-infected mice argues against differential activity of Akt. This may also apply to additional proteins, including the tumor suppressors p53 and PTEN. These pathways may contribute to the reduced number of small tumors in chemerin-156-overexpressing mice which is a subject of further investigations. The Fabp5 protein in HCC is associated with worse outcome and in vitro analysis revealed higher migration, invasion, and proliferation rates of cells with high Fabp5 protein levels [64]. Identical expression of Fabp5 protein in the tumors of both groups again indicated comparable malignancies.

Pnpla5 is a neutral lipid triacylglycerol lipase and, furthermore, has acylglycerol transacylase activity [45]. Pnpla5 mRNA was highly expressed in HCC tissues of chemerin-156-infected mice. Triglyceride and diacylglycerol species were nevertheless equally abundant in the tumors of both groups. This indicates that Pnpla5 is not a primary determinant of hepatic triglyceride and diacylglycerol levels. However, it is also possible that Pnpla5 protein and enzymatic activity were not induced in parallel to mRNA levels. It would be interesting to clarify the role of this enzyme in hepatic lipid metabolism. Elevated diacylglycerol concentrations contribute to HCC proliferation [65] and identical levels in the tumors of both animal groups further exclude major differences in HCC malignancy.

Normal expression of SREBP1c, FAS, and SCD1 showed that lipogenesis was not altered in the cancer tissues. HCC cells express low-density lipoprotein and very low-density lipoprotein receptors in abundance [66,67], and this enables the tumors to have a consistent supply of triglycerides and cholesterol. Concordantly, cholesterol concentrations were elevated in human HCC [68]. Cholesterol increases oxidative stress and DNA damage, and thus contributes to HCC growth [69]. In the present study, cholesterol was higher in the tumors of all animals. SREBP-2 is an important transcription factor enhancing cholesterol biosynthesis [69,70], but its active form was not changed in the tumors. ApoA1 protein was strongly diminished in the cancer tissues, and this illustrates that cholesterol accumulation may, in part, result from impaired reverse cholesterol transport [70].

In human HCC tissues, total chemerin protein declined in about 60% of the patients [14]. In the DEN model, chemerin protein was not changed in tumors of control-AAV injected mice. Chemerin mRNA expression was also similar in these tissues. Mice with chemerin-156 overexpression had comparable levels of total chemerin protein in tumorous and non-tumorous tissues. In a murine model of NASH-associated HCC, hepatic chemerin protein was unchanged in the tumors [47]. The described decline in chemerin protein in human HCC was not detected in murine HCC, and this is principally in accordance with normal chemerin protein levels in about 40% of human HCCs [14,47]. The antibody used to analyze chemerin by immunoblot detected all of the chemerin isoforms present in the liver. The question is whether chemerin variants differ in non-tumorous and tumorous tissues of mice and men.

Of note, chemerin mRNA expression strongly declined in the tumors of mice with chemerin-156 overexpression, though protein was not reduced. Chemerin mRNA and protein were not concordantly changed in epididymal fat of leptin receptor activity deficient db/db mice. Here, mRNA levels were normal and protein was raised about two-fold [71]. Chemerin mRNA expression may not correspond with protein levels. This was also the case with Fabp5 mRNA and protein, where only the former was found to be different in the tumor tissues between the two groups.

In human cohorts, high tumor chemerin was identified as a prognostic marker for survival [14]. The mechanisms involved in chemerin protein depletion in some cancers, chemerin isoform distribution, and the pathophysiological role in hepatocarcinogenesis needs further study.

Murine chemerin-156 and chemerin-155 are both highly active isoforms [27]. In the present study, chemerin-155 was the most abundant variant found in tumor tissues, whereas chemerin-156 was not detected. Chemerin-154 and chemerin-153, which are believed to be biologically inactive [27], were the two other isoforms found in liver cancers. Chemerin-153 was more abundant in the tumors of mice with chemerin-156 overexpression. Mast cell chymase cleaves chemerin-156 to generate chemerin-153 [4]. Interestingly, mast cell numbers were increased in HCC [72], and thus may have a role in processing active chemerin to inactive isoforms. Whether low chemerin protein in human HCC is really linked to worse survival because of the decline of biologically active and anti-carcinogenic chemerin isoforms requires further detailed analysis.

## 4. Materials and Methods

### 4.1. Adenoassociated Virus 8 (AAV8)

Murine chemerin cDNA to express chemerin-156 was cloned into the plasmid pAAV-AFP-MMAP-MCS. The mouse alpha-fetoprotein enhancer and the mouse minimal albumin promoter controlled the expression of the cDNA. Packaging plasmid was pDP8. AAV8 particles were produced in HEK293T cells and purified by iodixanol gradient centrifugation. Virus-expressing chemerin-156 was referred to as chemerin-156-AAV. AAV8 virus particles without cloned cDNA (control-AAV) served as control. The AAV8 particles were obtained from Sirion Biotech (Planegg-Martinsried, Germany) and were stored at −80 °C until use.

### 4.2. Animals

Male C3H/HeNRj mice were from Janvier Labs (Le Genest-Saint-Isle, France) and at 18–21 days of age were injected with 25 µg DEN (Sigma, Taufkirchen, Germany)/g body weight. DEN was dissolved in water. A total of 24 weeks later, chemerin-156-AAV or control-AAV (10^12^ virus per mouse) were intraperitoneally injected, and 13 weeks later (approximate age 39 weeks) the mice were euthanized by a CO_2_-caused coma, followed by cervical dislocation (Figure 1a). Macroscopically visible liver tumors were separated from non-tumorous tissue using a pair of binoculars [73]. Throughout the course of the study, mice were fed a standard chow (V1124-300, Mouse breading 10 mm autoclavable, Ssniff, Soest, Germany). Mice had free access to water and food and were housed in a 21 ± 1 °C controlled room under a 12 h light–dark cycle. All procedures were in accordance with the institutional and governmental regulations for animal use (Approval number 54-2532.1-21/14, 03,11,2014).

### 4.3. Sirius Red and Hematoxylin-Eosin Staining.

Sirius Red and hematoxylin-eosin staining was performed as previously described [47].

### 4.4. ELISAs

Chemerin ELISA was from R&D Systems (Wiesbaden-Nordenstadt, Germany). Mouse serum was diluted 1000-fold for chemerin analysis. ELISA to measure alpha-fetoprotein was from R&D Systems and serum was diluted 20-fold, as recommended.

### 4.5. Measurement of CMKLR1 and GPR1 Activity in Mouse Serum

Details of these assays were described elsewhere [74,75].

### 4.6. Mass Spectrometry of Chemerin Protein

Chemerin protein immunoprecipitated from the tumors was used for mass spectrometry. Protein was cut out from the gel and washed with 50 mM NH_4_HCO_3_, 50 mM NH_4_HCO_3_/acetonitrile (3/1), 50 mM NH_4_HCO_3_/acetonitrile (1/1), and lyophilized. After a reduction/alkylation treatment and additional washing steps, proteins were in gel digested with trypsin (Trypsin Gold, mass spectrometry grade, Promega, Mannheim, Germany) overnight at 37 °C. The resulting peptides were sequentially extracted with 50 mM NH_4_HCO_3_ and 50 mM NH_4_HCO_3_ in 50% acetonitrile. After lyophilization, peptides were reconstituted in 20 µL 1% trifluoroacetic acid and separated by reverse-phase chromatography. An UltiMate 3000 RSLCnano System (Thermo Fisher Scientific, Dreieich, Germany) equipped with a C18 Acclaim Pepmap100 preconcentration column (100 µm i.D. × 20 mm, Thermo Fisher Scientific) and an Acclaim Pepmap100 C18 nano column (75 µm i.d. × 250 mm, Thermo Fisher Scientific) was operated at a flow rate of 300 nL/min and a 60 min linear gradient of 4% to 40% acetonitrile in 0.1% formic acid. The liquid chromatographie was online-coupled to a maXis plus UHR-QTOF System (Bruker Daltonics, Leipzig, Germany) via a CaptiveSpray nanoflow electrospray source. Acquisition of mass spectrometry spectra after collision-induced dissociation fragmentation was performed in data-dependent mode at a resolution of 60,000. The precursor scan rate was 2 Hz, processing a mass range between *m*/*z* 175 and *m*/*z* 2000. A dynamic method with a fixed cycle time of 3 s was applied via the Compass 1.7 acquisition and processing software (Bruker Daltonics, Leipzig, Germany). Prior to database searching with Protein Scape 3.1.3 (Bruker Daltonics) connected to Mascot 2.5.1 (Matrix Science, London, UK), raw data were processed in Data Analysis 4.2 (Bruker Daltonics). A customized database comprising the *Mus musculus* entries from UniProt, as well as manually added sequences of the different chemerin processing forms and common contaminants, was used for database search with the following parameters: enzyme specificity trypsin with two missed cleavages allowed, precursor tolerance 10 ppm, MS/MS tolerance 0.04 Da. Deamidation of asparagine and glutamine, oxidation of methionine, carbamidomethylation, or propionamide modification of cysteine were set as variable modifications. The spectra of peptides corresponding to the C-terminus of the different chemerin processing forms were inspected manually.

### 4.7. Lipid Analysis

Lipid extraction was performed according to the method of Bligh and Dyer [76] in the presence of not naturally occurring lipid species as internal standards. Liver homogenates representing a wet weight of 2 mg were extracted. Chloroform phase was recovered by a pipetting robot (Tecan Genesis RSP 150, Zevenhuizen, Netherlands) and vacuum dried. The residues were dissolved either in 10 mM ammonium acetate in methanol/chloroform (3:1 *v/v*) (for low mass resolution tandem mass spectrometry) or chloroform/methanol/2-propanol (1:2:4 *v/v/v*) with 7.5 mM ammonium formate (for high resolution mass spectrometry). Lipid analysis was performed by direct flow injection analysis (FIA) using either a triple quadrupole mass spectrometer (FIA-MS/MS; low mass resolution setup as described previously [77]) or a hybrid quadrupole Orbitrap mass spectrometer (FTMS; high mass resolution) (QExactive, Thermo Fisher Scientific, Bremen, Germany). The Fourier transform mass spectrometry (FIA-FTMS) setup and data processing details are described in detail in Höring et al. [77].

### 4.8. Immunoblot

Protein was isolated with the AllPrep DNA/RNA/Protein Mini Kit (Qiagen, Hilden, Germany). The antibodies used, order number, dilution, and the respective companies are listed in Appendix A.

### 4.9. Semiquantitative Real-Time RT-PCR

RNA was isolated with the AllPrep DNA/RNA/Protein Mini Kit. RT-PCR was done as described in detail elsewhere [78]. Sequences of the primers are listed in Appendix A.

### 4.10. GeneChip Analysis

The Mouse Gene 2.1. ST Array (Affymetrix, Schwerte, Germany) was hybridized with RNA isolated from normal and tumorous liver tissues of control- and chemerin-156-infected mice (five animals per group). The Ambion WT Expression Kit and Affymetrix WT Terminal Labeling and Hybridization procedure were used according to the suppliers´ suggestions. Data were analyzed using the Affymetrix Command Console and Expression Console. Differences were calculated by the unpaired Student´s t-test (Kompetenzzentrum für Fluoreszente Bioanalytik, Regensburg, Germany). After Bonferroni correction, not a single gene was significantly changed in the tumor when compared to the respective non-tumorous tissues of control-AAV-infected animals. Real-time RT-PCR analysis revealed that Spink1 was significantly induced in the tumors and the respective *p*-value for this difference (*p* = 0.01289) was chosen as cut off value. Principle component analysis and cluster dendrogram were performed as described [79,80].

### 4.11. Recombinant Expression of Chemerin Isoforms in Hepa1–6 cells

Chemerin cDNA was amplified with the universe primer 5´- CGAAAGCTT ATGAAGTGCTTGCTGATCTCC -3‘and the reverse primers chemerin-162: 5´- CGA CCGCGGTTATTTGGTTCTCAGGGCCCTGGA-3´, chemerin-156: 5´- CGACCGCGG TTAGGAGAAGGCAAACTGTCCAGG -3´, chemerin-155: 5´- CGACCGCGGTTAGAA GGCAAACTGTCCAGGTAG -3´ or chemerin-154: 5´- CGACCGCGGTTAGGCAAACTG TCCAGGTAGGAA-3´ for cloning of chemerin-162, 156, 144, or 154, respectively, in the plasmid pcDNA3.1. The cleavage sites for the restriction endonucleases are underlined and all fragments were cloned with HindIII and SacII. The DNA-inserts were verified by sequencing (GeneArt, Regensburg, Germany).

### 4.12. Statistics

Data were displayed as box plots (median, lower, and upper quartiles and range of the values) or bar charts. Small circles indicate outliers greater than 1.5 times the interquartile range and stars indicate outliers greater than 3.0 times the interquartile range. Data of 9 control-AAV- and 12 chemerin-156-AAV-infected mice per group were given in the figures/tables unless stated otherwise. Statistical tests used were one-way ANOVA with the post-hoc Bonferroni test, Mann–Whitney U Test, and Spearman correlation (SPSS Statistics 25.0, IBM, Ehningen, Germany) and a value of *p* < 0.05 was regarded as significant.

## 5. Conclusions

In summary, the present study showed that chemerin-156 delayed early carcinogenesis but not late progression of HCC. Chemerin-155 was a prevalent isoform in liver tumors and future work is required to clarify the physiological and pathophysiological role of this variant.

## Figures and Tables

**Figure 1 ijms-21-00252-f001:**
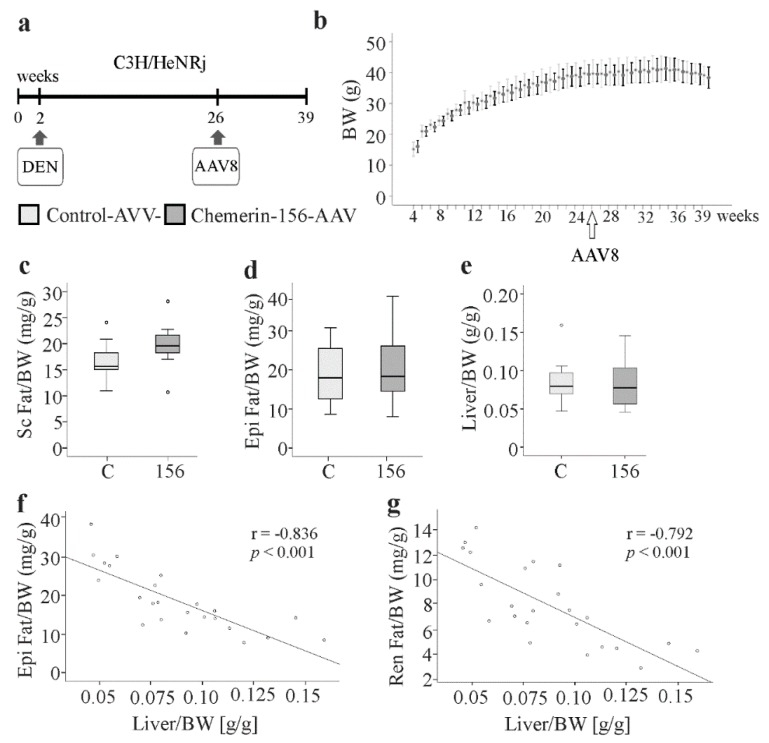
Experimental outline, body weight, and organ weights. (**a**) Experimental outline. (**b**) Body weight (BW) of control-AAV (adeno-associated virus) (C; *n* = 9) and chemerin-156 (156; *n* = 12)-AAV infected male mice during the study. Data are shown as mean ± standard deviation. (**c**) Subcutaneous (Sc) adipose tissue weight relative to BW. (**d**) Epididymal (Epi) adipose tissue weight relative to BW. (**e**) Liver weight relative to BW. (**f**) Correlation of Epi Fat/BW and liver/BW. (**g**) Correlation of perirenal (Ren) Fat/BW and liver/BW. Spearman correlation coefficient r and *p*-values are included in f and g. Small circles in c and e indicate outliers greater than 1.5 times the interquartile range.

**Figure 2 ijms-21-00252-f002:**
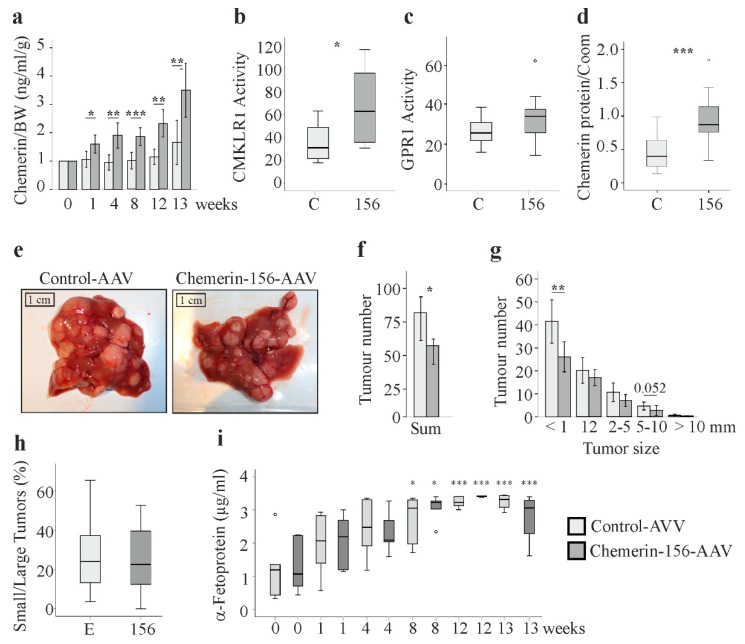
Chemerin protein, activity, tumor number, and α-fetoprotein. (**a**) Chemerin protein was analyzed by ELISA in serum of control-AAV (*n* = 9) and chemerin-156-AAV (*n* = 12) infected mice before and after AAV injection. (**b**) Serum activation of CMKLR1, given as a chemerin-156 equivalent in 9 mice injected with control-AAV and 12 mice injected with chemerin-156-AAV, as analyzed at the end of the study. (**c**) Serum activation of GPR1 of the animals, given as a chemerin-156 equivalent, as analyzed at the end of the study. (**d**) Chemerin protein in the liver of these animals. (**e**) Appearance of the livers. (**f**) Tumor number in the mouse livers. (**g**) Tumor number stratified for size. The number in the figure is the *p*-value for an almost significant difference. (**h**) Ratio of tumors with a diameter <2 mm to tumors with a diameter ≥2 mm. (**i**) Systemic α-fetoprotein at the time of AAV application (0 weeks) and 1, 4, 8, 12, and 13 weeks later. Data of five animals per group and time point are shown because there was not enough serum from all mice. Levels were significantly induced at 8, 12, and 13 weeks when compared to the respective control mice at 0 weeks in both groups. Small circles in d and i indicate outliers greater than 1.5 times the interquartile range. * *p* < 0.05, ** *p* < 0.01, *** *p* < 0.001.

**Figure 3 ijms-21-00252-f003:**
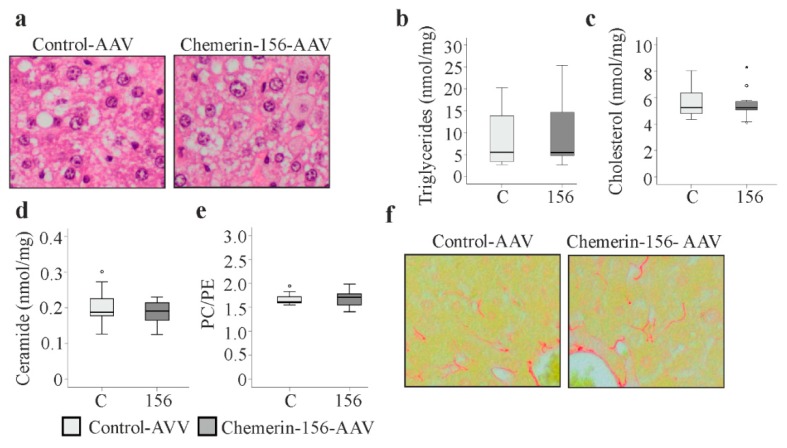
Analysis of hepatic injury in non-tumorous tissue of control-AAV and chemerin-156-AAV infected mice. (**a**) Hematoxylin and eosin stained liver. (**b**) Hepatic triglycerides. (**c**) Hepatic cholesterol levels. (**d**) Hepatic ceramide levels. (**e**) Hepatic phosphatidylcholine/phosphatidylethanolamine (PC/PE) ratio. (**f**) Sirius Red stained liver. Small circles in c, d and e indicate outliers greater than 1.5 times the interquartile range. The star in c indicates an outlier greater than 3.0 times the interquartile range.

**Figure 4 ijms-21-00252-f004:**
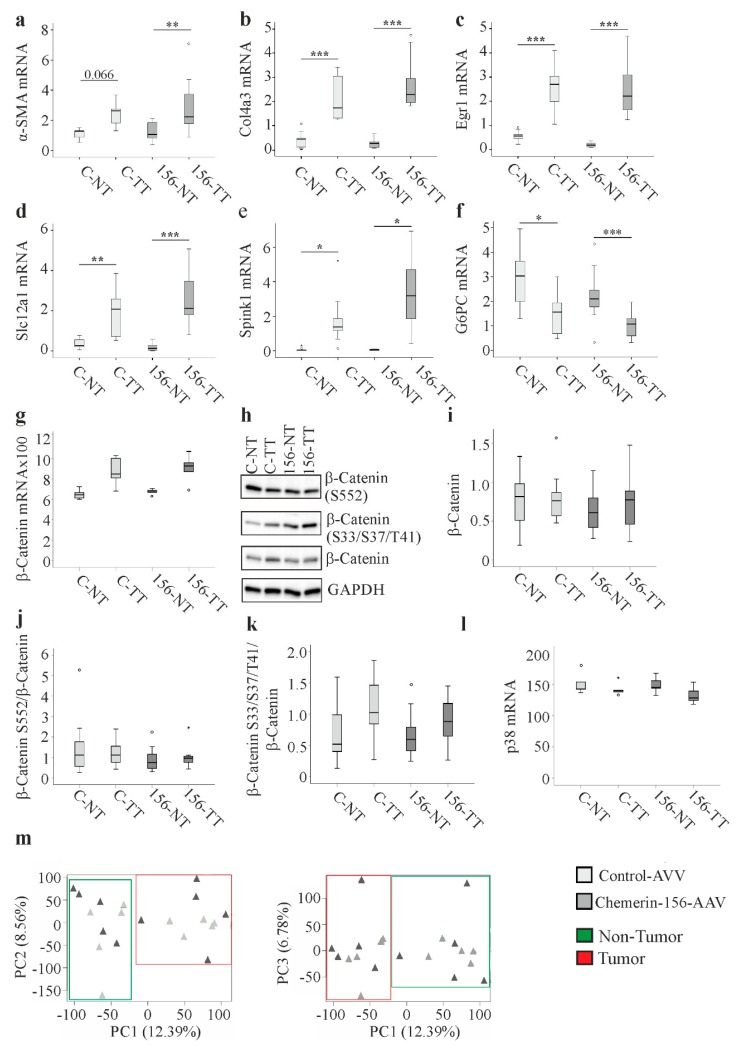
Principle component analysis of microarray data, and the expression of different genes and β-catenin proteins in hepatic non-tumorous (NT) and tumor tissue (TT) of control-AAV and chemerin-156-AAV infected mice. Data shown in g and l were obtained from GeneChip analysis, the expression of further genes was analyzed by real-time reverse-transcription polymerase chain reaction (RT-PCR). Expression of (**a**) *αSMA* (the number in the figure is the *p*-value for an almost significant difference). (**b**) *Col4a3*. (**c**) *Egr1*. (**d**) *Slc12a1*. (**e**) *Spink1*, and (**f**) *G6PC* mRNA. (**g**) *β-catenin* mRNA. (**h**) β-catenin and its phosphorylated forms. (**i**) Quantification of β-catenin protein. GAPDH was used for normalization. (**j**) Quantification of β-catenin protein phosphorylated at S552. Unphosphorylated β-catenin was used for normalization. (**k**) Quantification of β-catenin protein phosphorylated at S33, S37, or T41. Unphosphorylated β-catenin was used for normalization. (**l**) Expression of *p38* mRNA. (**m**) Principle component analysis of the microarray experiment where tumorous and non-tumorous tissues of control and chemerin-156-AAV infected mice were analyzed (*n* = 5 per group). Small circles in the figure indicate outliers greater than 1.5 times the interquartile range and small stars indicate outliers greater than 3.0 times the interquartile range. * *p* < 0.05, ** *p* < 0.01, *** *p* < 0.001.

**Figure 5 ijms-21-00252-f005:**
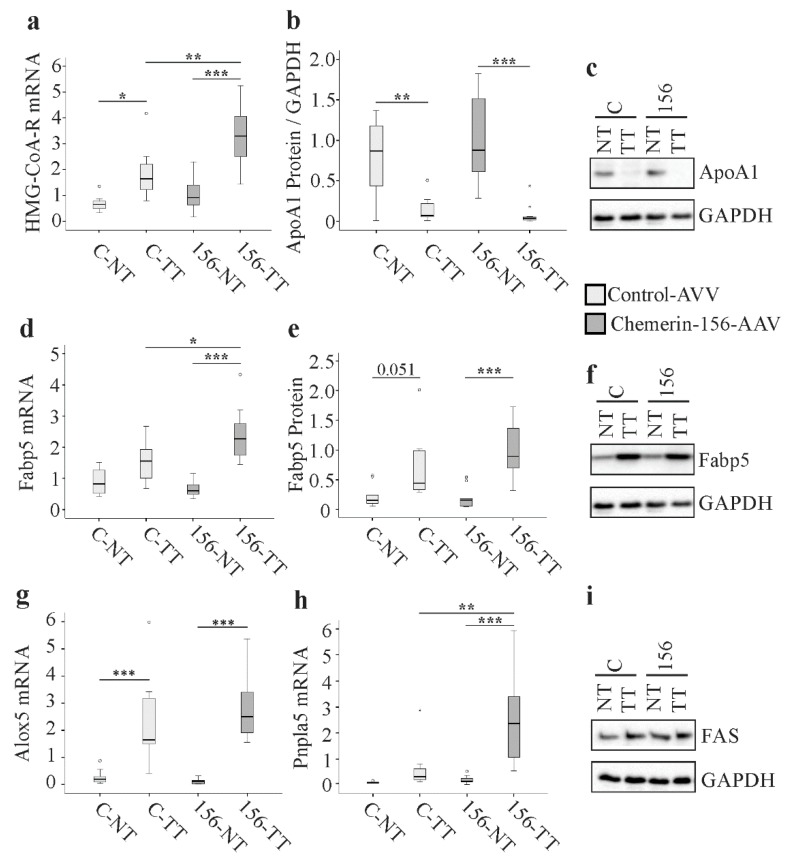
Levels of mRNA and protein for genes with a function in lipid metabolism in hepatic non-tumorous (NT) and tumor tissue (TT) of control-AAV (C) and chemerin-156-AAV (156) infected mice. (**a**) Expression of *HMG-CoA-R* mRNA. (**b**) Expression of ApoA1 protein. (**c**) Representative immunoblot of ApoA1 protein in NT and TT of both groups. (**d**) Expression of *Fabp5* mRNA. (**e**) Expression of Fabp5 protein. (**f**) Representative immunoblot of Fabp5 protein in NT and TT of both groups. (**g**) Expression of *Alox5* mRNA. (**h**) Expression of *Pnpla5* mRNA. (**i**) Representative immunoblot of FAS protein in NT and TT of both groups. Small circles in the figure indicate outliers greater than 1.5 times the interquartile range and small stars indicate outliers greater than 3.0 times the interquartile range. * *p* < 0.05, ** *p* < 0.01, *** *p* < 0.001.

**Figure 6 ijms-21-00252-f006:**
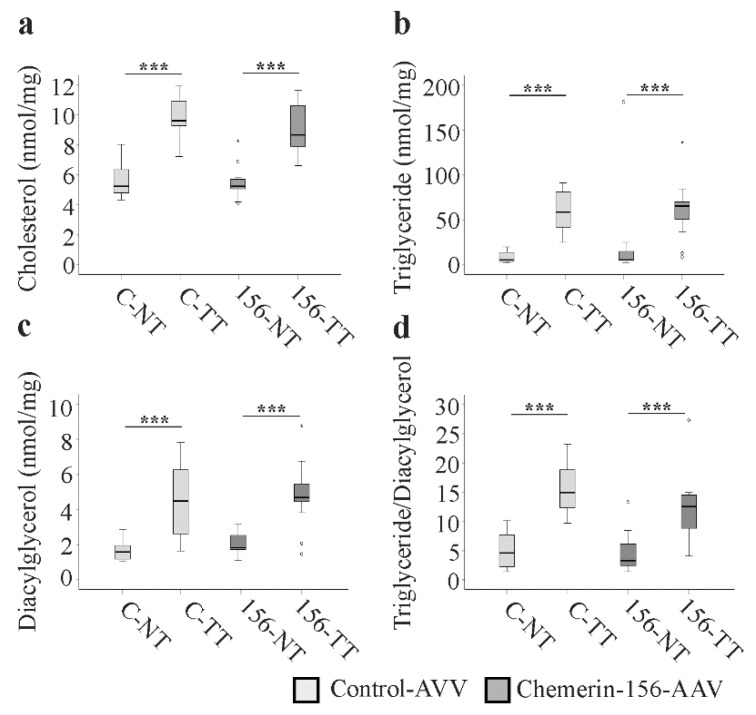
Lipids in hepatic non-tumorous (NT) and tumor tissue (TT) of control-AAV (C) and chemerin-156-AAV (156) infected mice. (**a**) Cholesterol. (**b**) Triglycerides. (**c**) Diacylglycerol. (**d**) Triglyceride to diacylglycerol ratio. Small circles in the figure indicate outliers greater than 1.5 times the interquartile range and small stars indicate outliers greater than 3.0 times the interquartile range. *** *p* < 0.001.

**Figure 7 ijms-21-00252-f007:**
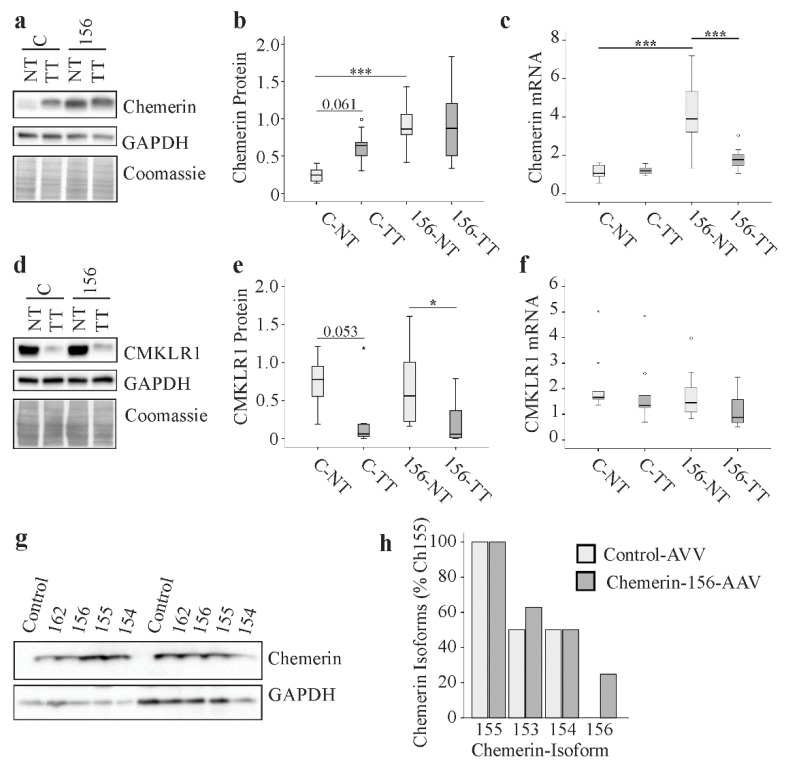
Chemerin and CMKLR1 in hepatic non-tumorous (NT) and tumor tissue (TT) of control-AAV (C) and chemerin-156-AAV (156) infected mice and chemerin isoforms in liver tumors. (**a**) Representative immunoblot of chemerin protein in the NT and TT of both groups. (**b**) Quantification of chemerin protein in NT and TT of control-AAV and chemerin-156-AAV infected mice. (**c**) *Chemerin* mRNA in the NT and TT of control-AAV and chemerin-156-AAV infected mice. (**d**) Representative immunoblot of CMKLR1 protein in NT and TT of both groups. (**e**) Quantification of CMKLR1 protein in the NT and TT of control-AAV and chemerin-156-AAV infected mice. (**f**) *CMKLR1* mRNA in the NT and TT of control-AAV and chemerin-156-AAV infected mice. (**g**) Chemerin in lysates of Hepa1–6 cells overexpressing chemerin-162, 156, 155, and 154. Data of two experiments are shown. (**h**) Chemerin isoforms in the liver tumors. Small circles in the figure indicate outliers greater than 1.5 times the interquartile range and small stars indicate outliers greater than 3.0 times the interquartile range. * *p* < 0.05, *** *p* < 0.001. Numbers in b and e are *p*-values for almost significant differences.

**Table 1 ijms-21-00252-t001:** Genes highly expressed in macrophages or natural killer cells were analyzed by real-time RT-PCR in the normal tissues (NT) and the tumor tissues (TT) of control-AAV and chemerin-156-AAV infected mice. Expression was not changed in either the tumors nor by chemerin-156 overexpression. Expression of CCL3 and IL-6 was not detected in tumor tissue of one mouse (most likely because of technical problems) and these values were not included in the calculation.

Gene	Animal Group	Tissue	Animal Number	Median and Range
*F4/80*	Control-AAV	NT	9	1.7 (1.1–6.1)
	Control-AAV	TT	9	2.1 (1.1–5.1)
	Chemerin-156-AAV	NT	12	1.4 (0.6–5.8)
	Chemerin-156-AAV	TT	12	1.2 (0.5–4.9)
*CCL3*	Control-AAV	NT	9	0.6 (0.2–0.9)
	Control-AAV	TT	8	0.4 (0.2–0.9)
	Chemerin-156-AAV	NT	12	0.5 (0.3–1.4)
	Chemerin-156-AAV	TT	12	0.3 (0.2 – 0.9)
*CD38*	Control-AAV	NT	9	1.5 (1.1–3.3)
	Control-AAV	TT	9	2.4 (1.8–4.1)
	Chemerin-156-AAV	NT	12	1.7 (0.5–3.3)
	Chemerin-156-AAV	TT	12	2.3 (2.0–3.1)
*IL-6*	Control-AAV	NT	9	1.7 (0.3–2.6)
	Control-AAV	TT	8	0.5 (0.2–2.4)
	Chemerin-156-AAV	NT	12	2.3 (0.2–7.2)
	Chemerin-156-AAV	TT	12	1.3 (0.1–3.5)
*NCR1*	Control-AAV	NT	9	1.6 (0.9–2.7)
	Control-AAV	TT	9	2.0 (1.3–3.4)
	Chemerin-156-AAV	NT	12	2.0 (0.6–4.9)
	Chemerin-156-AAV	TT	12	1.9 (0.9–5.1)
*Ly49c*	Control-AAV	NT	9	0.9 (0.6–1.6)
	Control-AAV	TT	9	0.7 (0.4–1.5)
	Chemerin-156-AAV	NT	12	0.8 (0.4–1.3)
	Chemerin-156-AAV	TT	12	0.5 (0.3–1.1)

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
