# Peer review of "Overexpression of Hepatocyte Chemerin-156 Lowers Tumor Burden in a Murine Model of Diethylnitrosamine-Induced Hepatocellular Carcinoma"

_ijms, 2019, doi:10.3390/ijms21010252_

Round 1

Reviewer 1 Report

In this manuscript, the authors investigated the anti-cancerous functions of Chemerin-156 (C156) in the course of hepatocellular carcinoma (HCC) development in diethylnitrosamine (DEN)-treated mice. They found that the number of small tumors was significantly decreased by C156-overexpressoin while that of large ones was not changed. They further investigated its underlying mechanism by transcriptomics, metabolomics, and biochemical approaches. However, these analyses revealed that the tumors of C156-overexpressed mice were not grossly different from those of control mice.

This manuscript is well written, and the experiments were well designed and performed. Results are robust and support their conclusion. However, the following points should be considered:

Alpha-SMA expression in tumor tissues could be from cancer associated fibroblasts. It is worth to mention this in the Discussion.

P38 mRNA expression is not a robust indicator for the activation of this kinase. The authors should perform WB for p38, and may also want to examine JNK (as well as ASK), ERK.

The authors should perform PCA or cluster analyses of microarray data to visualize the similarity.

The authors should discuss the mechanism of downregulation of CMKLR1 expression. Is that observed in a clinical setting? Is it possible that AAV8-mediated CMKLR1 overexpression prevents or suppresses HCC development in DEN-treated mice even after 21 weeks?

Author Response

We are very grateful to the reviewers for their helpful comments on our submitted paper. In the revised paper we added all of the data available at the moment. However, we could not provide new experimental data because of the short time admitted to submit a revised paper.

Reviewer 1

This manuscript is well written, and the experiments were well designed and performed. Results are robust and support their conclusion. However, the following points should be considered:

 Alpha-SMA expression in tumor tissues could be from cancer associated fibroblasts. It is worth to mention this in the Discussion.

This is now shortly addressed in the Discussion, page 13:

“ Cancer associated fibroblasts within the HCC environment contribute to disease progression. These cells express a-SMA which was associated with poor survival of patients with HCC. In the tumors a-SMA was comparably induced in both groups of mice in accordance with similar malignancy of liver tumors.”

P38 mRNA expression is not a robust indicator for the activation of this kinase. The authors should perform WB for p38, and may also want to examine JNK (as well as ASK), ERK.

We fully agree that further kinases may be analysed and that p38 phosphorylation has to be studied to get an idea on the activity of p38. Considering that different genes, proteins and lipid levels did not differ between the both groups it is, however, unlikely that there are any differences in central signalling pathways at least in the large tumors analyzed.

Analysis using western blots needs some time. Deadline for resubmission of the revised paper was the 24.12.19. Though the editor kindly prolonged this time until the 31.12. this was not helpful because in Germany most people are not at work between Christmas and New Year. Morover, we are not able to perform immunoblots during this time because we have to order the antibodies from the respective companies.

In case the reviewer is convinced that these further analysis will basically improve the paper we have to newly submit the paper to the journal because the experiments can not be performed within 1 or 2 weeks. We are not sure whether the paper will be reviewed by the identical reviewers.

The authors should perform PCA or cluster analyses of microarray data to visualize the similarity.

 Principle Component Analysis and Cluster Dendrogram analysis were done (please see figure 4 m and Supplementary Figure 1 which was included after the Supplementary Tables). It becomes clear that tumor and non-tumor tissues differ in gene expression. There were no differences between Control and Chemerin-156 infected mice.

The authors should discuss the mechanism of downregulation of CMKLR1 expression. Is that observed in a clinical setting? Is it possible that AAV8-mediated CMKLR1 overexpression prevents or suppresses HCC development in DEN-treated mice even after 21 weeks?

We added a paragraph in Discussion addressing this important issue.

“Chemerin expression was analyzed in different human cancers whereas levels of CMKLR1 were hardly investigated. The human protein atlas provides information about CMKLR1 in colorectal, breast, prostate, lung and thyroid cancer (https://www.proteinatlas.org). CMKLR1 protein was not prognostic in these different tumor diseases (https://www.proteinatlas.org). In the liver CMKLR1 is expressed by hepatocytes, Kupffer cells, bile-duct cells and hepatic stellate cells [54]. Preliminary data from our group suggest that CMKLR1 protein was reduced in non-viral human HCC (own unpublished data). Until now the factors regulating CMKLR1 in liver cells were hardly described. Adiponectin and IL-6 seem to induce CMKLR1 mRNA in hepatocytes. In hepatitis C infection hepatic CMKLR1 was normal, levels were reduced or induced in NASH [5]. Further research has to analyze CMKLR1 expression in HCC as well as chronic liver diseases. “    

Reviewer 2 Report

The study is very interesting, it might provide a reasonable therapeutically strategy on HCC treatment, especially on different stages of liver cancer progression.  However, it still needs some minor revision on the study and manuscript.

In the introduction part, could the author add a brief study and result of their experiments? About figure 1, did the author monitor the bodyweight of the mice when they did AAV injection to the mice? About figure 1 c and e, there is one mouse has huge variance, does the various come from the same mouse or different one? The study showed over-expression of chemerin-156 could lower the tumor burden, from figure 1e, the ratio of liver weight to body weight has variances, what's the reason for the variance, and did the author test early and later stage time point? And is there pathology difference among the small tumor livers and large tumor livers? About the figure 3a, did the author do H&E staining on different size tumors, since HCC is usually heterogeneity, and from the figure 2 showed there are some differences among different size tumors, though the liver weight to body weight ratio at the end of the experiments is no significant difference? From figure 4 the gene expression of the tumor tissue from the control and AAV-156 treated groups, have the author tried to do RNA-seq or increase the tested gene numbers to find some difference? Meanwhile, did the author do some IHC staining of beta-catenin expression in control and  AAV chemerin-156 treated mice liver tissues? If there is no difference on beta-catenin expression between control and chemerin-156 AAV treated groups, how about p53 expression, because many study showed beta-catenin activation is exclusive with P53 loss function in HCC. For 4.8 antibodies information, could the author add the detail information of them, such as the catalog number, dilution used in the study? To table 1, the expression of CCL3 and IL-6 tested on 8 samples, so the author should mention the different numbers for the study. At the same time, could the author mark the significant of tested markers in table 1? Did the author test chemerin-156 expression in some patients’ serum samples? And I found one of the author’s publications in 2019, maybe the author could mention it. About the study of the molecular mechanisms, there is one paper published in 2018 “Chemerin suppresses hepatocellular carcinoma metastasis through CMKLR1-PTEN-Akt axis”, since the author also studied CMKLR1, how about PTEN, AKT signal expression in the study? How about the inflammation markers staining, such as cd31, cd4/80? Maybe a little molecular mechanisms study could make the study better.

Round 2

Reviewer 1 Report

The authors had adequately addressed my comments.
I have checked the paper (ref 39) demonstrating that p38 activity was not changed in DEN-treated HCC compared to NT. I am looking forward to your future work on the mechanism underlying the tumor suppressive effects of Chemerin-156.